# Experimental Research on Improving Activity of Calcinated Coal Gangue via Increasing Calcium Content

**DOI:** 10.3390/ma16072705

**Published:** 2023-03-28

**Authors:** Yanpeng Zhao, Zhongzhe Zhang, Yongsheng Ji, Lei Song, Mingming Ma

**Affiliations:** 1Xuzhou Yinshan Track Concrete Co., Ltd., Xuzhou Metro Infrastructure Engineering Co., Ltd., Xuzhou Metro Group Co., Ltd., Xuzhou 221000, China; 2Jiangsu Key Laboratory Environmental Impact and Structural Safety in Engineering, China University of Mining and Technology, Xuzhou 221116, China; 3Jiangsu Collaborative Innovation Center for Building Energy Saving and Construct Technology, Jiangsu Vocational Institute of Architectural Technology, Xuzhou 221116, China

**Keywords:** coal gangue, mineral admixtures, calcination, adding calcium, activity evaluation

## Abstract

In this investigation, non-spontaneous combustion coal gangue was activated by two methods: (1) low-temperature calcination and (2) calcium addition. Differences in the activity of the activated coal gangue were studied at various calcination temperatures and amounts of calcium addition. Meanwhile, the cementation activity of the activated coal gangue was evaluated according to the activity effect analysis. Furthermore, the influences of the activated coal gangue on the cementation activity of cement were investigated. The results indicated that the activities of the activated coal gangue increased at a temperature between 500 °C and 700 °C. The calcium addition method can also increase the activity of coal gangue, with the effect being better when the gangue is mixed with slag. The addition of calcium and the calcination of coal gangue can promote the production of active minerals such as metakaolin, which is the main reason for the increased cementation activity.

## 1. Introduction

Air pollution in mining areas is becoming increasingly serious as open piles of coal gangue stones are exposed to weathering, producing dust and air pollution. The gangue dust may contain harmful elements, which, when inhaled, cause diseases and even cancer. The use of coal gangue to extract chemical products, recover beneficial minerals, and produce agricultural fertilizers and microbial fertilizers has been a recent focus of research. Coal gangue contains a large amount of alumina, which can be turned into aluminum, and carbon chemical products, which can be used to produce coal gangue cement, coal gangue bricks and blocks, and other building materials.

Coal gangue is a solid waste product of coal mining and washing and is currently one of the most abundant industrial solid wastes in China [1,2,3]. Coal gangue not only occupies large amounts of land and pollutes the environment, but is also prone to spontaneous combustion and explosions when stored in the open air [4,5,6]. Therefore, methods of recycling and utilizing coal gangue are urgently needed.

Spontaneous combustion coal gangue (SCCG) has a good reactivity and has been widely used as an active blending material in cement production [7,8,9,10,11]. However, SCCG resources are limited, and most of the coal gangue stored in mining areas are non-spontaneous combustion coal gangues (NSCCGs). Currently, there are two main ways to utilize NSCCGs [5,12,13,14,15,16]. The first is to use it as a raw material for cement production; however, its poor homogeneity makes it unsuitable, which limits the wide utilization of NSCCGs in cement production. The other utilization is as a concrete aggregate. However, when NSCCG is used to prepare concrete, water consumption is greatly increased and the concrete’s mechanical strength is also significantly reduced. In addition, when NSCCG is used as a concrete aggregate, the elemental carbon within NSCGG is not fully used, which is a waste of resources. As a consequence, NSCCG aggregates are currently only suitable for use in unimportant or temporary structures such as road base layers. Therefore, achieving high-quality resource utilization of NSCCGs is a pressing focus of global research.

Calcined coal gangue can achieve the same activity as SCCG, and thus, the calcination of coal gangue is an effective utilization method [17,18,19]. However, the reactivity of calcined coal gangue is still very low and when calcined coal gangue is used as a mineral admixture in concrete, the compressive strength of the concrete is lower compared with ordinary concrete. The main components of calcinated coal gangue are aluminosilicates, and the difference in the calcium content is the main reason for the different performances of the two concrete materials [20,21,22]. It may be possible to increase the compressive strength of concrete containing coal gangue by increasing the corresponding calcium content. Therefore, finding suitable methods to increase the calcium content of calcinated coal gangue and understanding the corresponding mechanisms of enhanced concrete compressive strength are significant for the development and application of coal gangue.

In this investigation, based on the removal of carbon and the utilization of its thermal energy, low-temperature calcination and calcium addition treatments were carried out to activate the reactivity of non-spontaneous combustion coal gangues. The activated coal gangue powders were then mixed with cement and their influence on the strength of cement mortar was studied. We also studied the activity of calcined and calcium-added coal gangue powder mixed with slag powders.

## 2. Materials and Methods

### 2.1. Materials

#### 2.1.1. Cement

Ordinary Portland cement P.O42.5R, produced by the Xuzhou Zhonglian Cement Group, Xuzhou, China, was used in the experiment. Its basic physical properties and chemical composition are shown in Table 1 and Table 2, respectively.

#### 2.1.2. Slag, Lime, and Desulfurized Gypsum

In this investigation, lime and anhydrous calcium sulfate were used to add calcium to coal gangue. A mixture of slag and calcium-added coal gangue was prepared. The chemical composition of the slag, lime, desulfurization gypsum, and calcined coal gangue powders is shown in Table 2.

### 2.2. Preparation of Active Coal Gangue Mineral Admixture

#### Crushing and Grinding

The black non-spontaneous coal gangue used in this investigation came from the Datun Coal Mine in Xuzhou. The chemical composition of the coal gangue after combustion is shown in Table 2. Large lumps of coal gangue were first crushed to prepare particles with diameters < 0.3 mm using a jaw crusher. Then, pure non-spontaneous coal gangue and coal gangue with additions of 10% lime and 5% desulfurization gypsum were separately ground using a ball mill to prepare pure non-spontaneous coal gangue powders and calcium-added non-spontaneous coal gangue powders with a specific surface area of about 450 m^2^/kg.

The pure non-spontaneous coal gangue powders and calcium-added non-spontaneous coal gangue powders were each divided into three parts and then calcined in a muffle furnace at temperatures of 500 °C, 600 °C, and 700 °C to study the effect of calcination temperature on the activity of the coal gangue.

### 2.3. Experimental Contents and Schemes

#### 2.3.1. Effects of Calcination and Calcium Addition on Coal Gangue Activity

Samples of P.O 42.5R cement were prepared containing pure calcined coal gangue powders or calcium-added coal gangue powders after calcination at mass fractions of 0%, 10%, 30%, 50%, and 70%. The flexural and compressive strengths of the specimens were determined according to the “Test Method for Strength of Cement Mortar (ISO Method)”, GB/T17671-1999 [23], to compare the effects of the coal gangue content, calcium addition, and calcination on the activity of calcined coal gangue powders.

#### 2.3.2. Performances of Calcined and Calcium-Added Coal Gangue Powders Mixed with Slag

Slag was blended with calcium-added coal gangue powders after calcination at 600 °C at proportions of 0%, 10%, 20%, 30%, 40%, and 50%. Then, the prepared mixtures were used to replace 30% of the cement, and the flexural and compressive strengths of the cement specimens were determined according to the standard method for cement mortar; in addition, the activity was also analyzed.

#### 2.3.3. Analysis of the Physical Composition of Calcium-Enhanced Gangue

A.XRD

XRD was used for mineral composition analysis and to study the evolution of the chemical composition of the calcined coal gangue powder (CG), the calcium-enhanced coal gangue powder (CGA), and calcined calcium-enhanced coal gangue powder and slag compound (CGAS).

XRD measurements were performed using a D8 ADVANCE model (BRUKER equipment). The powdered samples were scanned between 5°and 65° with a 2 min increment of 0.02° per step, and Cu Ka radiation was applied.

B.XRF

To study the evolution of the the chemical compositions of the calcined coal gangue powder (CG), calcium-enhanced coal gangue powder (CGA), and calcined calcium-enhanced coal gangue powder and slag compound (CGAS), XRF was used for the analysis of the composition of the substances. The experiment was carried out using a wavelength dispersive X-ray fluorescence spectrometer, a BRUKER S8 TIGER model, from Bruker AXS (Billerica, MA, USA), for the quantitative and qualitative analysis of the test samples.

## 3. Experimental Results and Analyses

### 3.1. Effect of Calcination Temperature on the Activity of Pure Coal Gangue Powders

The compressive strengths of cement mortar containing calcined pure coal gangue powders are shown in Figure 1. The figure shows that at a given replacement ratio t, the compressive strength increases with an increasing temperature, but the differences are not significant. At a given temperature, the early strength changes are significant, while the later compressive strengths are not obviously reduced. Meanwhile, the development of flexural strength is promoted at coal gangue powder contents of <10%. With further increases in powder content, the early strengths of the cement mortar are significantly reduced.

### 3.2. Effect of Calcination Temperature on the Activity of Calcium-Containing Coal Gangue

The compressive strengths of mortar with calcined calcium-added coal gangue powders is shown in Figure 2. This figure shows that, at a fixed powder content, the 3-day compressive strengths of mortar calcined at different temperatures vary greatly, while the 28-day compressive strengths are similar.

Basically, the compressive strength increases as the calcination temperature of the calcium-containing coal gangue powders increases. At a calcination temperature of 500 °C, the 3-day compressive strengths of the powders are the lowest. At a constant calcination temperature, the changes in compressive strength are identical in mortars mixed with calcined coal gangue powders with and without calcium, but the changes in flexural strength are slightly different. The 28-day flexural strengths significantly decrease at a content of 10%. With further increases in the dosage, changes in the flexural strength are not obvious. When the content exceeds 50%, the flexural strength significantly decreases.

Figure 3 shows the mechanical strengths of the mortars with the two activated coal gangue powders. Figure 3c shows that when the temperature is constant, the strengths of both activated coal gangue powders decrease as the substitution amount decreases. The strengths of mortars with calcium-containing powders are higher than those with pure powders, and differ according to the substitution amount. This indicates that the activity of coal gangue powders can be significantly improved by calcination or calcium activation.

### 3.3. Effect of Calcined Calcium-Containing Coal Gangue Powder and Slag on Cement Mortar Strength

Figure 4 shows the compressive strengths of coal gangue–slag cement mortar and pure cement mortar. The figure shows that for the cement with a 30% coal gangue–slag mixture, the peak strength can be obtained with a 20% slag content. This strength is close to that of the pure cement mortar.

At slag contents < 20%, the 28-day and 60-day compressive strength curves are flat, and both are lower than those of pure cement mortar. With an increasing slag proportion, the mortar strength gradually increases, and peaks were formed at slag proportions of 30–40%, where the strength is greater than that of pure cement mortar. The strengths gradually decrease and tend to stabilize with further increases in slag proportions.

### 3.4. Evaluation of Calcined Active Coal Gangue

#### 3.4.1. Activity Evaluation Method

Based on the strength method, Pu Xincheng [24] proposed strength indicators such as the specific strength, specific strength coefficient, and strength contribution rate. Because the compositions of non-calcined coal gangues are relatively complex, the activity of active coal gangue powders was evaluated according to volcanic ash activity, as proposed by Pu Xincheng [24]:(1)$$\varphi =\frac{R_{t}}{R_{0}}$$
(2)$$\varphi =\frac{R_{t}}{R_{0}}$$
(3)$$\Psi =\frac{R_{t}-R_{0}}{R_{t}}\times 100\%$$

R_t_—Volcanic ash specific intensity;

φ—Volcanic ash specific intensity coefficient;

Ψ—Intensity contribution rate of volcanic ash effect (%);

C—The strength value of the active blend (MPa);

d—Percentage of cement usage (%);

R_0_—Specific strength of non-blended cement.

In general, the activity of the mineral admixtures is high if the specific strength coefficient is high. If the specific strength coefficient is higher than one, it indicates that there is greater strength development in the cementitious material with an admixture added than in that without an admixture added. The admixture has an active effect and promotes cement hydration. The strength of the mortar specimens at each age comes from the cement and admixture. Since the cement contents of the mortar specimens in each test group are different, the strength cannot be used directly to evaluate the admixture activity. Therefore, a strength contribution rate is proposed to represent the admixture activity. Different contents of admixtures contribute different amounts of strength to mortar specimens. If the strength contribution rate is positive, it indicates that the admixture contributes more strengths, while negative rates indicate a small strength contribution. Both the specific strength coefficient and strength contribution rate can be used to evaluate the activity of admixtures.

#### 3.4.2. Evaluation of the Activity of Calcined Pure Coal Gangue Powders

The specific strength coefficient and strength contribution rate of calcined pure coal gangue powders were calculated according to Formulas (1) to (3), and the results are illustrated in Figure 5.

From Figure 5a, it can be seen that with a 10% dosage of coal gangue powders calcined at 500 °C, 600 °C, and 700 °C, the early specific strength ratio is relatively low and is close to or less than 1, which indicates that the admixture is unable to promote the early hydration of cement and that the admixture has low activity. Furthermore, with admixture contents > 10%, the specific strength ratio gradually decreases to <1, indicating that increasing the admixture hinders the early hydration of cement. The 28-day specific strength ratio of calcined pure coal gangue powders is generally > 1 and increases as the admixture content increases, indicating that the admixture promotes later strength development in the cement, and that this effect is stronger at higher contents.

These strength changes in the mortar specimens can also be observed from the strength contribution rates shown in Figure 5b. As shown in Figure 5b, the strength contribution rate of the cement mortar containing 10% calcined coal gangue powder is close to zero at 3 days. With increasing contents, the ratio of the strength contribution of the admixture to the early strength of the cement mortar is negative, indicating that the calcined coal gangue powders restrain strength development in the cement mortar. The strength contribution rate in the later stage is generally greater than zero, and increases as the content increases, indicating that the admixture can improve the later strength of the cement mortar. These results are consistent with those based on the specific strength.

#### 3.4.3. Evaluation of the Activity of Calcined Calcium-Containing Gangue Powders

Figure 6 shows the compressive strength ratio and compressive strength contribution rate of calcined calcium-containing gangue powders. As is shown in Figure 6a, the early strength ratios of calcium- containing gangue powders calcined at different temperatures are close to or greater than one, and are generally higher than those of pure calcined coal gangue powders, indicating that the calcium-containing powders can promote the strength of mortar specimens. These increases are obvious in calcium-containing powder contents of 30% and 50%. The late strength ratios at 28 d and 60 d are >1 and gradually increase as the content of calcium-containing powder increases. Hence, calcined calcium-containing powders demonstrate late strength development due to cement hydration in the later period. This effect is significantly better than those of calcined pure coal gangue powders. The above changes in compressive strength can also be obtained from the compressive strength contribution rates shown in Figure 6b.

#### 3.4.4. Evaluation of Activity of Calcined Calcium-Containing Coal Gangue Powders Mixed with Slag

Figure 7 shows variations in the compressive strength ratio and compressive strength contribution rate of calcined calcium-containing coal gangue powders mixed with slag. From Figure 7a,b it can be seen that when the slag content of the mixtures is within 20%, the 3-day early compressive strength ratios and strength contribution rates of calcined calcium-containing coal gangue powders mixed with slag increase with the increase in slag contents. Both peak at a slag content of 20%, which indicates that the calcium-containing powder is capable of increasing the early strength of cement when the slag content is low. When the slag content is >20%, the mixtures are still able to promote early strength development, but the effect gradually decreases with the increase in slag content.

At slag contents of 30–40%, the later compressive strength ratios and strength contribution rates of the mixtures are relatively high. As shown in Figure 7, at slag contents < 40%, the later strength contributions of the mixtures increase with the increase in slag content, although not significantly. Meanwhile, at slag contents > 40%, the later strength contributions of the mixtures decrease alongside the increase in slag content. This indicates that the calcium-containing powders with slag make a weak contribution to early strength development and a significant contribution to later strength development.

### 3.5. Analysis of the Physical Composition of Calcium-Enhanced Gangue

#### 3.5.1. XRF Analysis

It can be seen from the XRF data (Table 3) that the CO_2_ and SO_3_ contents of the untreated gangue were much higher than those of the calcined and modified treatments, which resulted in untreated gangue having a larger burn loss. The burn loss is mainly composed of crystalline water, C, S, and other volatile elements. In the hydration process, C, S, and other volatile elements in the cementation material do not contribute much to bond hardening. The gangue activity is mainly dependent on Ca, Si, Al, and other active elements. The lower the contents of non-active elements, including C, S, and other volatile element contents, the higher the activity of the gangue. The modifying treatment has a significant effect on the addition of calcium to the gangue: untreated coal gangue < CG < CGA < CGAS.

#### 3.5.2. XRD Analysis

From the X-ray diffractogram (Figure 8) of the gangues modified in different ways, it can be seen that the uncalcined and calcium-enhanced gangues mainly contain: quartz, kaolinite, muscovite, and illite.

The peak intensity of the quartz structure in the calcined gangue did not change significantly. The kaolinite diffraction peaks decreased in height, which was mainly due to the high-temperature fusion of kaolinite that formed amorphous SiO_2_ and Al_2_O_3_. The peak intensities of muscovite and illite were not significantly affected; therefore, they were not the aluminosilicate compounds.

After calcination at a high temperature, the diffractive matter and peak height of the gangue with calcium increased, while those of the gangue mixed with mineral powder did not change significantly. This means that the modification treatment did not affect the crystal structure of the gangue after calcination. However, AFt and AFm diffraction peaks appeared in the diffraction spectrum of the modified gangue, indicating that new material production can be promoted by modifying the gangue.

## 4. Mechanism Analysis

The chemical compositions of coal gangue are similar to those of clay. Generally, the higher the contents of silicon dioxide and aluminum oxide in coal gangue, the higher its activity after calcination. Since the coal content in non-spontaneous coal gangue is relatively high, carbon has a significant impact on the strength, water consumption, and durability of the resulting cement. Therefore, in this investigation, non-spontaneous coal gangue was calcined to remove carbon and thermally activate the gangue. Through calcination, the kaolinite component in the gangue was dehydrated and decomposed at a certain temperature, generating metakaolin, amorphous silicon dioxide, and aluminum oxide, according to the following process:550~700 °C: Al_2_O_3_·2SiO_2_·2H_2_O → Al_2_O_3_·2SiO_2_ + 2H_2_O
Al_2_O_3_·2SiO_2_
→ Al_2_O_3_ + 2SiO_2_

The non-spontaneous combustion coal gangue was chemically activated by adding calcium-containing lime and desulfurization gypsum, then mixing and calcining it with slag. During the calcination process, amorphous silica and alumina were formed, and chemical reactions generated strength in the presence of CaO, CaSO_4_, and water, according to the following process:Al_2_O_3_ + 3CaO + 3CaSO_4_ + 32H_2_O → 3CaO·Al_2_O_3_·3CaSO_4_·32H_2_O
Al_2_O_3_ + 3CaO + CaSO_4_ + 18H_2_O → 3CaO·Al_2_O_3_·CaSO_4_·18H_2_O
Al_2_O_3_ + 4CaO + 13H_2_O → 4CaO·Al_2_O_3_·13H_2_O
SiO_2_ + CaO + xH_2_O → CaO·SiO_2_·xH_2_O

However, at a high addition amount, the activity of the calcined coal gangue powders was much lower than that of pure cement, indicating that calcium addition and calcination have a limited capacity to improve the activity of NSCCG. As the contents of the activated coal gangue powders increased, the increased activity due to calcium addition was much weaker than the decrease effect on mortar strengths.

## 5. Conclusions

The activity of coal gangue powders can be improved by both low-temperature calcination and calcium-contained calcination, with the latter being more effective. The activity of calcined coal gangue powders increases at temperatures in the range of 500–700 °C.When calcined coal gangue powder is used as a cement admixture at a proportion of <10%, changes in the later strength are not significant, but these changes become more obvious at higher contents.By mixing low-temperature (600 °C) calcined and calcium-added coal gangue powders with slag and using them to replace 30% of the cement, the later strengths at 28 d and 60 d are higher than those of pure cement with admixture slag contents of 30–40%. The mixtures can replace pure cement, but are disadvantageous in terms of early strength development at 3 d. Therefore, these mixtures are suitable for the production of cement with low early strength requirements.

## Figures and Tables

**Figure 1 materials-16-02705-f001:**
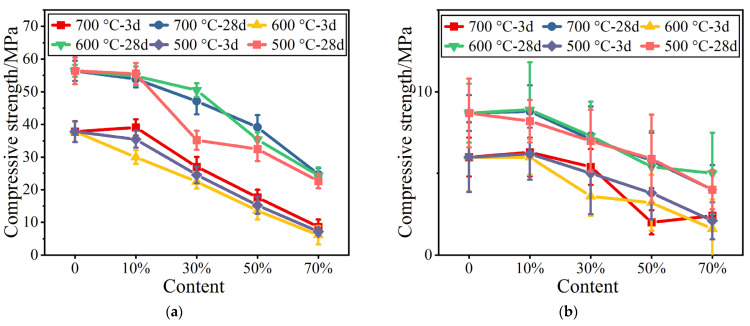
Mechanical strengths of calcined coal gangue. (**a**) Compressive strength. (**b**) Flexural strength.

**Figure 2 materials-16-02705-f002:**
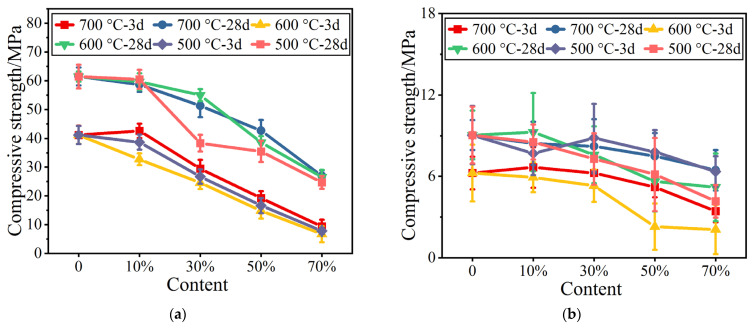
Mechanical strengths of calcium-added calcined gangue. (**a**) Compressive strength. (**b**) Flexural strength.

**Figure 3 materials-16-02705-f003:**
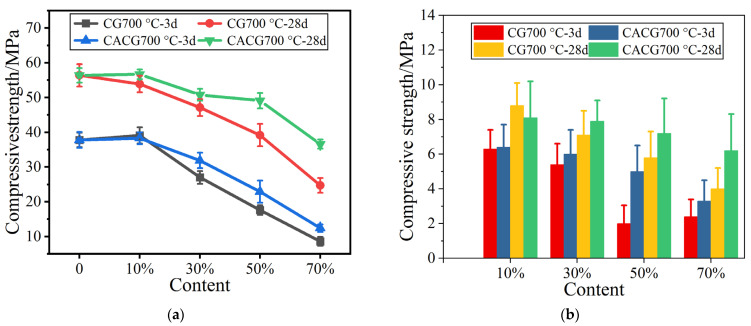
Strengths of mortars with calcined gangue and calcium-added gangue. (**a**) Compressive strength after 700 °C. (**b**) Flexural strength after 700 °C. (**c**) Compressive strength after 600 °C. (**d**) Flexural strength after 600 °C. (**e**) Compressive strength after 500 °C. (**f**) Flexural strength after 500 °C.

**Figure 4 materials-16-02705-f004:**
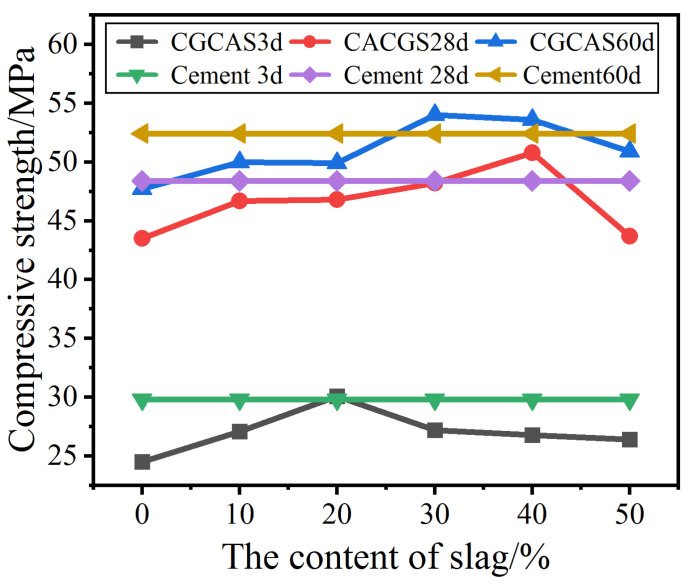
Compressive strengths of mortars added with gangue–slag admixture.

**Figure 5 materials-16-02705-f005:**
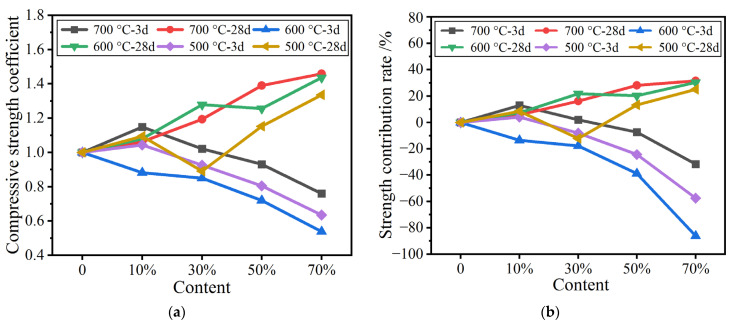
(**a**) Specific compressive strength coefficient; (**b**) Strength contribution rate.

**Figure 6 materials-16-02705-f006:**
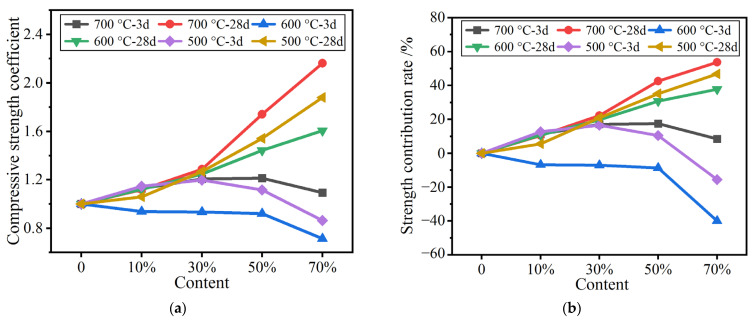
(**a**) Specific compressive strength coefficient for mortar with calcium-added gangue (**b**) Strength contribution rate of mortar with calcium-added gangue.

**Figure 7 materials-16-02705-f007:**
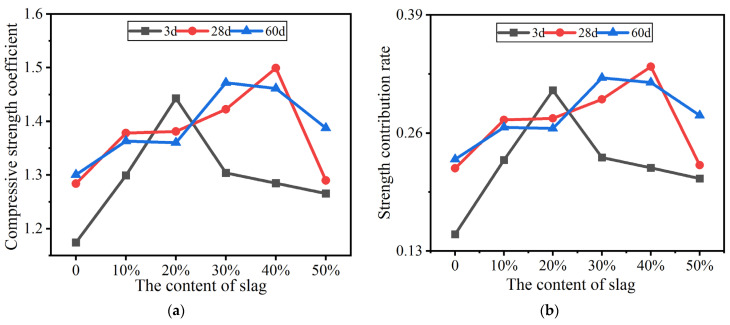
(**a**) Specific compressive strength coefficient of gangue–slag admixture; (**b**) Strength contribution rate of gangue–slag admixture.

**Figure 8 materials-16-02705-f008:**
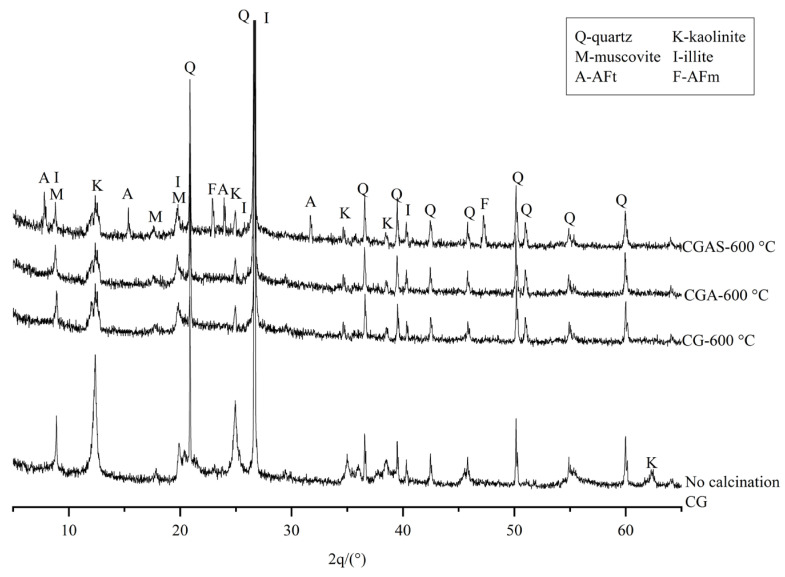
XRD of coal gangue with different treatment methods.

**Table 1 materials-16-02705-t001:** Physical properties of cement.

	Standard Consistency Water Consumption/%	Fineness of (80 μm) Square-Hole Sieve Residue/%	Setting Time (min)		Flexural Strength/MPa		Compressive Strength/MPa	
			Initial Setting	Final Setting	3 d	28 d	3 d	28 d
P.O.42.5	28	1.8	195	320	4.97	7.15	27.6	48.3

**Table 2 materials-16-02705-t002:** Chemical composition of materials (wt%).

	SiO_2_	A1_2_O_3_	Fe_2_O_3_	CaO	SO_3_	MgO	Na_2_O	Ti_2_O	K_2_O	Loss
Cement	26.55	7.77	3.62	54.59	2.24	2.68	0.31	-	1.50	3.2
Slag	31.35	18.65	0.57	34.65	-	9.31	-	-	-	-
Lime	16.98	13.35	7.43	30.29	-	1.5	2.82	2.29	-	-
Desulfurized gypsum	2.7	0.7	0.5	31.6	42.4	1.0	-	-	-	19.2
Calcined coal gangue powder	61.24	18.50	2.58	1.48	0.61	0.52	0.14	-	1.53	13.41

**Table 3 materials-16-02705-t003:** XRF of different treatment methods for gangue.

Chemical Composition	Untreated Coal Gangue	Calcined Coal Gangue (CG)	Calcium-Enhanced Coal Gangue Powder (CGA)	Calcined Calcium-Added Gangue Powder and Slag Compound (CGAS)
CO_2_	12.38	6.21	6.11	6.01
SO_3_	4.23	2.23	2.12	2.06
SiO_2_	59.12	62.02	63.21	63.67
Al_2_O_3_	15.32	18.43	16.25	16.15
Fe_2_O_3_	2.71	2.75	2.56	2.34
CaO	0.5	1.57	4.76	7.59
TiO_2_	0.04	0.05	0.06	0.07
MgO	0.12	0.48	0.54	0.61

## Data Availability

Not applicable.

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
