# Peer review of "Experimental Research on Improving Activity of Calcinated Coal Gangue via Increasing Calcium Content"

_materials, 2023, doi:10.3390/ma16072705_

Round 1

Reviewer 1 Report

in principle; The research idea is good and falls within the scope journal

But

- It is well known that the samples used in the work are either samples that have been previously analyzed and the author is attributed to the reference, or that the authors analyze them; And then you must write the name of the used instrument. We note that the authors do not  write the names of the devices that were relied upon in the analyzes, especially the XRF.

- Important note to the authors; The best article relies on three basic elements:

1- good idea

2- Good discussion

3- Good and clear output.

So; Authors must make all forms using advanced programs such as Origin, Sigma Plot, or Excel, but with professionalism. in drawing and transportation.

- Although the author used cost-effective waste materials, if they did mention the economics of the product, they did not refer to the economic importance and the environmental dimension in the article.

Author Response

Remark 1: - It is well known that the samples used in the work are either samples that have been previously analyzed and the author is attributed to the reference, or that the authors analyze them; And then you must write the name of the used instrument. We note that the authors do not write the names of the devices that were relied upon in the analyzes, especially the XRF.

Response: Thank you for your careful advice.

The experimental analysis results of XRD and XRF have been supplemented in the article, and the names of equipment and instruments have been added in the second part.

Table 1 XRF of different treatment methods for gangue

Chemical composition

Untreated coal gangue

Calcined coal gangue(CG)

Calcium-enhanced coal gangue powder(CGA)

Calcined calcium-added gangue powder and slag compound (CGAS)

CO2

12.38

   6.21

6.11

6.01

SO3

4.23

   2.23

2.12

2.06

SiO2

59.12

62.02

63.21

63.67

Al2O3

15.32

18.43

16.25

16.15

Fe2O3

2.71

2.75

2.56

2.34

CaO

0.5

1.57

4.76

7.59

TiO2

0.04

0.05

0.06

0.07

MgO

0.12

0.48

0.54

0.61

           Figure 1 XRD of coal gangue with different treatment methods

Remark 2: - Important note to the authors; The best article relies on three basic elements:

1- good idea; 2- Good discussion; 3- Good and clear output.

So; Authors must make all forms using advanced programs such as Origin, Sigma Plot, or Excel, but with professionalism. in drawing and transportation.

Response: Thank you for your careful advice. We have redrawn all the data diagrams in this article

Remark 3:  Although the author used cost-effective waste materials, if they did mention the economics of the product, they did not refer to the economic importance and the environmental dimension in the article.

Response: Thank you for your careful advice. We have added the economic importance and the environmental dimension to the introduction.

With the development and utilization of coal, air pollution in mining areas is becoming more and more serious. A large number of coal gangue stones piled up in the open air will produce dust due to weathering, causing air pollution.Because coal gangue dust may contain elements that are harmful to the human body, it will be inhaled into the lungs by the human body through breathing, causing diseases in the body, and more seriously, it can also cause cancer.The use of coal gangue to extract chemical products, recover beneficial minerals, and produce agricultural fertilizers and microbial fertilizers has also been greatly developed.At the same time, a large amount of alumina contained in coal research stone can be used to prepare aluminum-based and carbon-based chemical products, and to produce building materials such as coal gangue cement, coal gangue bricks and blocks.

Reviewer 2 Report

The search for additional cementing materials (DCM) is an urgent task. The authors activate the coal waste rock by firing at low temperatures. Activated carbon gangue powder is then used as a partial replacement for cement as a mineral additive. The study investigates the effect of the content of coal gangue powder on the strength of the cement slurry, and also studies the activity of the mixed powder of calcined coal gangue with the addition of calcium and slag powder.

Remarks

1.         The goal is to remove carbon and use its thermal energy to activate its activity, however, nothing is written about this in the study.

2.         In figures 1-3, you must specify the confidence interval.

3.         Figures 4 and 5 require further explanation.

Requires an explanation due to which an increase in strength is achieved with a slag content of 30 to 40% (Fig. 4)

Needs an explanation why the gangue coal powder has low activity at an early stage. Why is its activity manifested in the later stages, and the greater the dosage, the greater the stimulating effect (Fig. 5)

4.         The reaction (line 263) is doubtful. The article does not provide evidence of the formation of oxides during the decomposition of kaolin.

5.         The formation of AFt and AFm phases (line 269, 270 and 271) directly from their oxides is doubtful and without evidence.

Conclusion

The scientific article may be of interest to researchers dealing with the problem of using additional cementing materials (ACM).

The article can be recommended for publication with the revision of the comments made.

Author Response

Remark 1: The goal is to remove carbon and use its thermal energy to activate its activity, however, nothing is written about this in the study.

Response: Thank you for your careful advice.

The XRF( as shown in Table 1) shows that the CO2 and SO3 content of the untreated gangue is much higher than that of the calcined and modified treatment, resulting in a larger burn loss in the untreated gangue. The main losses in the burn loss are the water of crystallisation, C, S and other volatile elements, which contribute less to the bond hardening of the cementitious material during the hydration process C, S and other volatile elements. Therefore the activity size of the gangue mainly depends on Ca, Si, Al and other active elements, and the smaller the content of non-active elements C, S and other volatile elements, the higher the activity of the gangue.

Table 1 XRF of different treatment methods for gangue

Chemical composition

Untreated coal gangue

Calcined coal gangue(CG)

Calcium-enhanced coal gangue powder(CGA)

Calcined calcium-added gangue powder and slag compound (CGAS)

CO2

12.38

   6.21

6.11

6.01

SO3

4.23

   2.23

2.12

2.06

SiO2

59.12

62.02

63.21

63.67

Al2O3

15.32

18.43

16.25

16.15

Fe2O3

2.71

2.75

2.56

2.34

CaO

0.5

1.57

4.76

7.59

TiO2

0.04

0.05

0.06

0.07

MgO

0.12

0.48

0.54

0.61

Remark 2:  In figures 1-3, you must specify the confidence interval.

Response: Thank you for your careful advice. We have added a confidence interval in In figures 1-3 as required

(a) Compressive strength                    (b)Flexural strength

Fig.1 Mortar strength of gangue for calcined

(a) Compressive strength                    (b)Flexural strength

Fig.2 Mortar strength of increased calcium gangue for calcined

(a) Compressive strength of 700℃            (b) Flexural strength of 700℃

(c) Compressive strength of 600℃            (d) Flexural strength of 600℃

(e) Compressive strength of 500℃            (f) Flexural strength of 500℃

Fig.3 Strength comparison between gangue calcined and increased calcium gangue

Remark 3:  Figures 4 and 5 require further explanation. Requires an explanation due to which an increase in strength is achieved with a slag content of 30 to 40% (Fig. 4)Needs an explanation why the gangue coal powder has low activity at an early stage. Why is its activity manifested in the later stages, and the greater the dosage, the greater the stimulating effect (Fig. 5)

Response: Thank you for your careful advice. Because coal gangue and other types of mineral blends require a certain hydration reaction time, the fuller the hydration time, the more its mechanical strength develops and the more perfect the internal structure, the secondary hydration reaction of mineral blends such as coal gangue promotes the conversion of Ca (OH)2 to generate more CSH gels, the main strength component, which leads to the later stage. The more coal gangue is incorporated, the stronger the reaction in the later stage.

Remark 4:  The reaction (line 263) is doubtful. The article does not provide evidence of the formation of oxides during the decomposition of kaolin.

Response: Thank you for your careful advice.

通过XRD图中可以看到未煅烧的煤矸石中高岭石的衍射峰高度明显小于改性煤矸石的衍射峰高度,由于同种物质的质量含量是一定的,满足物质守恒定律,因此经过煅烧后的煤矸石中高岭石发生了部分转化。通过对比可以看到XRF图中,未煅烧煤矸石中含有的Al2O3和SiO2,从XRF分析的表格中可以看出,未煅烧的煤矸石中Al2O3和SiO2的含量分别为15.32 59.12

Through the XRD diagram(as shown in Figure1), it can be seen that the height of the kaolinite in the uncalcined coal gangue is significantly smaller than the height of the kaolinite in the modified coal gangue. Since the mass content of the same substance is certain, it satisfies the law of conservation of matter, so the kaolinite in the calcined coal gangue has undergone partial transformation. Through comparison, we can see the Al2O3 and SiO2 contained in the uncalcined coal gangue in the XRF diagram (as shown in Table 1). From the table analyzed by the XRF, it can be seen that the content of Al2O3 and SiO2 in the uncalcined coal gangue is 15.32, 59.12, respectively.

The content of Al2O3 and SiO2 in the calcined coal gangue has increased significantly, and the activity of coal gangue is mainly determined by its internal Al2O3 and SiO2. Based on the above analysis, the following formula is obtained. The following formula shows the process of this reaction, and although the following reaction can occur, it is not 100% conversion.

           Figure 1 XRD of coal gangue with different treatment methods

Remark 5:   The formation of AFt and AFm phases (line 269, 270 and 271) directly from their oxides is doubtful and without evidence.

Response: Thank you for your careful advice.

The occurrence of the following reactions is based on the data of the XRD reaction(as shown in Figure 1), and the occurrence of the following reactions is not a 100% conversion, but is based on experimental data and the conditions under which the chemical reaction occurs. For example, the secondary hydration reaction of mineral blends consumes Ca (OH)2 to generate CSH. Although this reaction occurs, it is not a 100% conversion.

Al2O3+3CaO+3CaSO4+32H2O→3CaO·Al2O3·3CaSO4·32H2O

Al2O3+3CaO+CaSO4+18H2O→3CaO·Al2O3·CaSO4·18H2O

Reviewer 3 Report

The work is related to the utilization of Coal gangue as a solid waste product obtained during the process of coal mining. The work is of interest to civil engineers and specialists in the field of ecology, circular economy and building materials. Classical methods were used to track the physical-mechanical properties (Compressive strength, Flexural strength) of new building materials.
I have the following remarks about the work:
1. The paragraph between lines 67-69 is redundant.
2. What is meant by "5% desulfurization gypsum" (line 84). As it is written, does it mean CaO? What substance is added - is it CaO or anhydrous CaSO4
3. Table 2:
The authors should check the values of the indicators in table 2. There is a discrepancy between the data for CaO and SO3 - about 9% with the indicated data. A content of 31.6% CaO should correspond to a value of SO3 approximately 45%.
4. Fig. 3, c, d, f contains inscriptions that are not in English.
5. The presented mechanism of several reactions - lines 262, 263, 269-272 should be supported by a powder phase analysis.
6. It is need to specify the conditions and temperatures at which the reactions take place.

Author Response

Remark 1: The paragraph between lines 67-69 is redundant.

Response: Thank you for your careful advice. We have deleted the paragraphs between lines 67-69.

Remark 2: What is meant by "5% desulfurization gypsum" (line 84). As it is written, does it mean CaO? What substance is added - is it CaO or anhydrous CaSO4

Response: Thank you for your careful advice. Desulfurized gypsum is CaSO4. I am very sorry for the ambiguity caused to you by my wrong expression. The added material is gypsum.

Remark 3: Table 2:The authors should check the values of the indicators in table 2. There is a discrepancy between the data for CaO and SO3 - about 9% with the indicated data. A content of 31.6% CaO should correspond to a value of SO3 approximately 45%.

Response: Thank you for your careful advice. We have re-checked Table 2. Because the raw materials contain certain impurities, this phenomenon occurs.

Remark 4: Fig. 3, c, d, f contains inscriptions that are not in English.

Response: Thank you for your careful advice. We have modified Figures 3 c-f.

Remark 5: The presented mechanism of several reactions - lines 262, 263, 269-272 should be supported by a powder phase analysis

Response: Thank you for your careful advice. We have supplemented the data of XRD and XRF to support the reaction mechanism

Table 1 XRF of different treatment methods for gangue

Chemical composition

Untreated coal gangue

Calcined coal gangue(CG)

Calcium-enhanced coal gangue powder(CGA)

Calcined calcium-added gangue powder and slag compound (CGAS)

CO2

12.38

   6.21

6.11

6.01

SO3

4.23

   2.23

2.12

2.06

SiO2

59.12

62.02

63.21

63.67

Al2O3

15.32

18.43

16.25

16.15

Fe2O3

2.71

2.75

2.56

2.34

CaO

0.5

1.57

4.76

7.59

TiO2

0.04

0.05

0.06

0.07

MgO

0.12

0.48

0.54

0.61

           Figure 1 XRD of coal gangue with different treatment methods

Remark 6:. It is need to specify the conditions and temperatures at which the reactions take place.

The conditions and temperature at which the reaction occurs have been supplemented, which are at room temperature (20℃).

Round 2

Reviewer 1 Report

Authors do the required modifications and notes.

Author Response

Authors do the required modifications and notes.

Response: Since we did not mark the reply comments in the comments of the previous round of replies, we are very sorry for this. We will now highlight the first round of review comments in the text, and attach the revised content to the comments

Round 1 Comment

The work is related to the utilization of Coal gangue as a solid waste product obtained during the process of coal mining. The work is of interest to civil engineers and specialists in the field of ecology, circular economy and building materials. Classical methods were used to track the physical-mechanical properties (Compressive strength, Flexural strength) of new building materials.

Remark 1: - It is well known that the samples used in the work are either samples that have been previously analyzed and the author is attributed to the reference, or that the authors analyze them; And then you must write the name of the used instrument. We note that the authors do not write the names of the devices that were relied upon in the analyzes, especially the XRF.

Response: Thank you for your careful advice.

(1) The experimental analysis results of XRD and XRF have been supplemented in the article, and the names of equipment and instruments have been added in the 2.3(3). The specific content is as follows(The corresponding position in the manuscript has been highlighted):

(3) Analysis of the physical composition of calcium-enhanced gangue

  1. XRD

In order to study the evolution of the chemical composition of calcined coal gangue powder (CG), calcium-enhanced coal gangue powder (CGA) and calcined calci-um-enhanced coal gangue powder and slag compound (CGAS), XRD was used for min-eral composition analysis.

XRD measurement was performed by BRUKER equipment (model D8 ADVANCE). The powdered samples were scanned between 5°and 65° with a 2 min increment 0.02° per step, and a Cu Ka radiation were applied.

  1. XRF

In order to study the evolution of the chemical composition of calcined coal gangue powder (CG), calcium-enhanced coal gangue powder (CGA) and calcined calci-um-enhanced coal gangue powder and slag compound (CGAS), XRF was used for the analysis of the composition of the substances. The experiment was carried out using a wavelength dispersive X-ray fluorescence spectrometer, model BRUKER S8 TIGER, from Bruker AXS, Germany, for the quantitative and qualitative analysis of the test samples.

(2) We have added the analysis content of XRD and XRF in section 3.5 of this article, as follows:

3.5 Analysis of the physical composition of calcium-enhanced gangue

  1. XRF analysis

It can be seen from the XRF (Table 3) that the CO2 and SO3 content of the untreated gangue is much higher than that of the calcined and modified treatment, which results in a larger burn loss of the untreated gangue. The loss in the burn loss is mainly crystalline water, C, S and other volatile elements, in the hydration process C, S and other volatile elements for the cementation material of the bond hardening help less. So the gangue activity size mainly depends on Ca, Si, Al and other active elements, and non-active elements C, S and other volatile elements content is smaller, the higher the activity of the gangue. The modifying treatment has a significant effect on the calcium addition to the gangue.: Untreated coal gangue < CG<CGA<CGAS.

Table 1 XRF of different treatment methods for gangue

Chemical composition

Untreated coal gangue

Calcined coal gangue(CG)

Calcium-enhanced coal gangue powder(CGA)

Calcined calcium-added gangue powder and slag compound (CGAS)

CO2

12.38

   6.21

6.11

6.01

SO3

4.23

   2.23

2.12

2.06

SiO2

59.12

62.02

63.21

63.67

Al2O3

15.32

18.43

16.25

16.15

Fe2O3

2.71

2.75

2.56

2.34

CaO

0.5

1.57

4.76

7.59

TiO2

0.04

0.05

0.06

0.07

MgO

0.12

0.48

0.54

0.61

  1. XRD analysis

From the X-ray diffractogram (Figure 8) of the gangue modified in different ways, it can be seen that the uncalcined and calcium-enhanced gangue contains mainly: quartz, kaolinite, muscovite  and illite.

The peak intensity of the quartz structure in the calcined gangue did not change significantly. kaolinite diffraction peaks decreased in height, mainly due to the high temperature fusion of kaolinite to form amorphous SiO2 and Al2O3. The peak intensities of muscovite and illite were not significantly affected and therefore the aluminosilicate compounds were not significantly affected.

The diffractive matter and peak height of the gangue with calcium increase and the gangue mixed with mineral powder did not change significantly after calcination at high temperature, which means that the gangue after modification treatment did not affect the crystal structure of the gangue after calcination. However, diffraction peaks of AFt and Afm appear in the diffraction peaks of the modified gangue, indicating that new material production can be promoted by modifying the gangue.

           Figure 8 XRD of coal gangue with different treatment methods

Remark 2: - Important note to the authors; The best article relies on three basic elements:

1- good idea; 2- Good discussion; 3- Good and clear output.

So; Authors must make all forms using advanced programs such as Origin, Sigma Plot, or Excel, but with professionalism. in drawing and transportation.

Response: Thank you for your careful advice. We have redrawn all the data diagrams in this article(The title of the data has been highlighted). The data graph is shown below:

(a) Compressive strength                    (b)Flexural strength

Fig.1 Mechanical strengths of calcined coal gangue

(a) Compressive strength                    (b)Flexural strength

Fig.2 Mechanical strengths of calcium-added calcined gangue

(a) Compressive strength after 700℃            (b) Flexural strength after 700℃

(c) Compressive strength after 600℃            (d) Flexural strength after 600℃

(e) Compressive strength after 500℃            (f) Flexural strength after 500℃

Fig.3 Strengths of mortars respectively with calcined gangue and calcium-added gangue

Fig4 Compressive strengths of mortars added with gangue - slag admixture

  Fig5(a) Specific compressive strength coefficient   Fig5(b) Strength contribution rate

  \

Fig6(a) Specific compressive strength coefficient    Fig6(b) Strength contribution rate of mortar

for mortar with calcium-added gangue         with calcium-added gangue

Fig7(a) Specific compressive strength coefficient      Fig7(b) Strength contribution rate of gangue

of gangue - slag admixture                          - slag admixture

Remark 3:  Although the author used cost-effective waste materials, if they did mention the economics of the product, they did not refer to the economic importance and the environmental dimension in the article.

Response: Thank you for your careful advice. We have added the economic importance and the environmental dimension to the introduction of the first paragraph. The corresponding content added is as follows(The corresponding position in the manuscript has been highlighted):

With the development and utilization of coal, air pollution in mining areas is becoming more and more serious. A large number of coal gangue stones piled up in the open air will produce dust due to weathering, causing air pollution. Because coal gangue dust may contain elements that are harmful to the human body, it will be inhaled into the lungs by the human body through breathing, causing diseases in the body, and more seriously, it can also cause cancer. The use of coal gangue to extract chemical products, recover beneficial minerals, and produce agricultural fertilizers and microbial fertilizers has also been greatly developed. At the same time, a large amount of alumina contained in coal research stone can be used to prepare aluminum-based and carbon-based chemical products, and to produce building materials such as coal gangue cement, coal gangue bricks and blocks.

We have tried our best to revise and improve the manuscript and made the changes in the manuscript according to the Reviewer’s good comments. We appreciate for Editors/Reviewer’ warm work earnestly, and hope that the corrections will meet with approval. Once again, we acknowledge your comments and constructive suggestions very much, which are valuable in improving the quality of our manuscript.

If there are other errors or further requests, please contact us by e-mail.

Yours sincerely,

Yongsheng Ji

Reviewer 3 Report

The authors have improved the quality of their publication by taking into account the comments made by the reviewer.
The work is now more precise, but the ambiguity of the term "desulfurization gypsum" remains. In the responses, the authors indicate that it is anhydrous calcium sulfate. In the chemical literature, it is accepted to use the name "anhydrite" or "anhydrous calcium sulfate" for anhydrous calcium sulfate. The term desulfurization gypsum is not only inaccurate but also very wrong, because gypsum means CaSO4.2H2O, and desulfurization means sulfur-free. Gypsum without crystal water means anhydrite, and CaSO4 without sulfur - CaO.
In the sense that "desulfurization gypsum" is used in work, the term must be changed. I believe it is appropriate to use "anhydrous calcium sulfate".
To be published after correction of incorrect terms.

Author Response

The authors have improved the quality of their publication by taking into account the comments made by the reviewer.

The work is now more precise, but the ambiguity of the term "desulfurization gypsum" remains. In the responses, the authors indicate that it is anhydrous calcium sulfate. In the chemical literature, it is accepted to use the name "anhydrite" or "anhydrous calcium sulfate" for anhydrous calcium sulfate. The term desulfurization gypsum is not only inaccurate but also very wrong, because gypsum means CaSO4.2H2O, and desulfurization means sulfur-free. Gypsum without crystal water means anhydrite, and CaSO4 without sulfur - CaO.

In the sense that "desulfurization gypsum" is used in work, the term must be changed. I believe it is appropriate to use "anhydrous calcium sulfate".

To be published after correction of incorrect terms

Response: Thank you very much for your affirmation of the other comments we revised in the first round. We are very sorry that our explanation of "desulfurization gypsum" is not accurate enough. We have changed the terminology according to your suggestion, all using "anhydrous calcium sulfate"
